# Virulence and Genetic Diversity of *Puccinia* spp., Causal Agents of Rust on Switchgrass (*Panicum virgatum* L.) in the USA

**DOI:** 10.3390/pathogens14020194

**Published:** 2025-02-14

**Authors:** Bochra A. Bahri, Peng Tian, Samikshya Rijal, Katrien M. Devos, Jeffrey L. Bennetzen, Shavannor M. Smith

**Affiliations:** 1Department of Plant Pathology, University of Georgia, Athens, GA 30602, USA; 2Institute of Plant Breeding, Genetics and Genomics, University of Georgia, Athens, GA 30602, USA; s.rijal@uga.edu (S.R.); kdevos@uga.edu (K.M.D.); 3The Division of Plant Science and Technology, University of Missouri, Columbia, MO 65201, USA; tianp@missouri.edu; 4Department of Crop and Soil Sciences, University of Georgia, Athens, GA 30602, USA; 5Department of Plant Biology, University of Georgia, Athens, GA 30602, USA; 6Department of Genetics, University of Georgia, Athens, GA 30602, USA; maize@uga.edu

**Keywords:** switchgrass, rust, virulence, genetic diversity

## Abstract

Switchgrass (*Panicum virgatum* L.) is an important cellulosic biofuel grass native to North America. Rust, caused by *Puccinia* spp. is the most predominant disease of switchgrass and has the potential to impact biomass conversion. In this study, virulence patterns were determined on a set of 38 switchgrass genotypes for 14 single-spore rust isolates from 14 field samples collected in seven states. Single nucleotide polymorphism (SNP) variation was also assessed in 720 sequenced cloned amplicons representing 654 base pairs of the elongation factor 1-α gene from the field samples. Five major haplotypes were identified differing by 11 out of the 39 SNP positions identified. STRUCTURE, Principal Coordinate Analysis, and phylogenetic analyses divided the rust population into two genetic clusters. Virginia and Georgia had the highest and lowest rust genetic diversity, respectively. Only nine accessions showed a differential disease response between the 14 isolates, allowing the identification of eight races, differing by 1–3 virulence factors. Overall, the results suggested clonal reproduction of the pathogen and a North–South differentiation via local adaptation. However, similar haplotypes and races were also recovered from several states, suggesting migration events, and highlighting the need to further investigate the switchgrass rust population structure and evolution in the USA.

## 1. Introduction

Switchgrass (*Panicum virgatum*) is a polyploid and perennial C4 grass that is native to North America. It is widely adapted to drought, cold, and heat. Switchgrass primarily outcrosses and exists in different ecotypes, including lowland and upland. Lowland accessions are mostly tetraploid and upland accessions are octoploid, and the two ecotypes vary in their morphology and regions of adaptation [1]. Switchgrass was traditionally used for forage and soil conservation in the Midwest and the Great Plains. However, in 1991, it was selected by the Department of Energy (DOE) as a model herbaceous bioenergy crop due to its high biomass yield potential and its ability to be cultivated on marginal lands with low nutrient and water inputs [2]. It was previously predicted that plant pathogens would have minimal effects on switchgrass yield losses because it is a native species. However, reports of various diseases affecting bioenergy crops are increasing [3,4,5,6,7,8,9,10]. In fact, switchgrass has been found to serve as a host to 174 different pathogenic and non-pathogenic fungi [10], so far. In Georgia, rust infections were observed each year on many accessions of the Southern Switchgrass Diversity Panel (SSDP) during the period 2011–2019, which was maintained in Watkinsville, GA [11]. Foliar diseases caused by rust fungal pathogens have caused yield losses impacting switchgrass production [12,13]. Once the crop is grown in monoculture over larger acreages, switchgrass biomass production is expected to be severely impacted by rust epidemics.

Six species of Puccinia have been reported to cause rust in switchgrass: *P. emaculata*, *P. graminicola* (syn. Uromyces graminicola), *P. pammelii*, *P. amari*, *P. novopanici*, and *P. pascua* [14,15]. Two of the species, *P. graminicola* and *P. emaculata*, have been widely reported in the USA. [10]. The predominant species was *P. emaculata*, as documented across Georgia, Arkansas, Tennessee, Alabama, Mississippi, Iowa, Virginia, Oklahoma, West Virginia, Texas, Missouri, Kansas, Nebraska, Michigan, New York, Pennsylvania, and South Dakota [11,16,17,18,19,20,21,22]. *P. emaculata* is now considered to not be present on switchgrass [14,15]. *P. novopanici* was reported as predominant in the Eastern and Central U.S., while *P. graminicola* and *P. pammelii* were more frequent in the Midwest [14,15,21]. Rust species can be differentiated into two groups based on morphology: *P. emaculata*, *P. novopanici*, *P. pammelii*, and *P. amari* are characterized by 2-celled teliospores, while *P. graminicola* and *P. pascua* have single-celled teliospores [14]. Species-specific variation in rust fungi was first determined based on rDNA sequencing, including partial Internal Transcribed Spacer [ITS], Intergenic Spacer [IGS], and part of the Large Ribosomal Subunit [LSU] [14,15,21,23,24]. ITS2 and IGS sequences were more informative in species differentiation compared to LSU [14,15]. *P. graminicola* and *P. pascua* were revealed to be closely related, forming a distinct genetic clade [15]. The other three species, *P. novopanici*, *P. pammelii*, and *P. emaculata*, were distinct and formed their own genetic clade. The development of diagnostic assays for species identification has not been entirely successful. Primers specifically amplifying *P. emaculata*, SGR-SP1-FW, and SGR-SP1-RV, were developed from ITS sequences [18]. These primers, however, were later shown to amplify an additional two species, *P. novopanici* and *P. amari*. Further examination of the primer sequences showed them to be designed for *P. novopanici*, suggesting that the previously reported isolates of *P. emaculata* were likely misidentified [14]. Recently, genetic variations within *P. novopanici* were detected using a whole genome sequencing approach, and a population genetic analysis revealed two distinct genetic subpopulations (northern and southern) across 88 samples [23]. Advances in molecular tools would provide a deeper understanding of the population genetics and diversity of rust pathogens within and between species.

Evaluation of germplasm for rust and various other disease responses, coupled with genetic association analysis, has led to the identification of accessions carrying genes conferring resistance to the invading pathogen and the development of disease-resistant cultivars in maize, wheat, rice, and other crops [25,26,27,28,29]. To facilitate switchgrass research and breeding, several populations were developed and studied, including diversity panels and crossed populations (F_1_ and half-sib) [30,31,32,33]. Most of these studies focused on assessing diverse physiological traits, including plant architecture, leaf wax, lignin and pectin biosynthesis, and biotic resistance, and identifying molecular markers associated with these traits. Differential responses to rust have been observed in commercially important upland and lowland cultivars [12,18]. However, limited research has been conducted to elucidate the genetic basis of resistance to rust in switchgrass. A quantitative trait locus (QTL) mapping study conducted on 431 F_2_ progeny derived from a four-way cross involving AP13, DAC6, WBC, and VS16 identified two major QTLs, *Prr1* on chromosome 3N and *Prr2* on chromosome 9N, and several minor QTLs associated with rust resistance [21]. The study suggested a strong genotype-by-environment interaction and differential host responses potentially explained by different rust species across geographic locations. Later, a genome-wide association study on a diversity panel of 630 switchgrass accessions evaluated across 10 locations in the central-northern U.S. and central-southern U.S. identified significant SNPs for rust resistance underlying *Prr1* and *Prr2* [23]. The candidate gene for *Prr1* was *Pavir.3NG168388*, which encodes an oligopeptide transporter. The significant SNP in *Prr2* was associated with *Pavir.9NG474500* (CCR4-NOT complex), which is known for the regulation of gene expression [23].

In addition, the genetic diversity and population structure of resistance gene homologs (RGHs) were previously analyzed across a wide range of switchgrass germplasm for four NBS-LRR RGH families [34]. The results demonstrated that the analyzed RGHs were under positive selection in the switchgrass accessions and exhibited variable recombination frequencies, which should generate new resistance genes with new specificities that could provide resistance to new races that develop in the pathogen population. A comprehensive analysis of pathogen variation with respect to pathogenicity and the correlation of sequence variation with the genetic determinants of the host resistance phenotype is needed.

In this study, we assessed switchgrass rust isolates collected in seven states in the U.S. for (1) their virulence patterns based on a set of 38 switchgrass genotypes maximizing genetic diversity from the SSDP, and (2) their genetic diversity based on SNP variations in the elongation factor 1-α (*EF-1*α) gene. Overall, the results suggested the population structure and clonal reproduction of switchgrass rust, and North–South local adaptation despite the diversity in haplotype, race, and gene flows between the investigated states.

## 2. Materials and Methods

### 2.1. Sampling Procedure and Establishment of a Puccinia *spp.* Isolate Collection from Switchgrass Fields

Two *Puccinia* spp. field isolates were collected from symptomatic switchgrass genotypes in Alabama, Georgia, Oklahoma, Tennessee, Texas, Virginia, and Wisconsin for a total of 14 field samples (Table 1). Symptomatic switchgrass leaf samples were selected for collection of *Puccinia* spp. spores based on the observation of high sporulation (susceptibility) on their leaves. Notations were made of the location of the collection in the field and of the switchgrass cultivar on which the infection developed. Switchgrass leaf samples were evaluated in the field and then brought to the lab where *Puccinia* spp. spores were collected from a single plant using a vacuum spore collector (University of Minnesota, St. Paul, MN, USA). For species identification, DNA was extracted from the spore samples and amplified with Internal transcribed spacer (ITS) primers [35], confirming the samples as *Puccinia* spp. and belonging to the *Puccinia novopanici*/*emaculata* genetic group (Appendix A Appendix A). Additionally, Koch’s Postulate was performed, and leaf samples were sent to the Diagnostics Laboratory at the University of Georgia to verify the identification of *Puccinia* spp. based on morphology.

### 2.2. Puccinia *spp.* Single Spore Isolation

Urediniospores from each field isolate (fourteen isolates) were mixed with talcum powder (3MgO-4SiO_2_-H_2_O; 1:3 ratio) in an Eppendorf tube and rubbed separately onto the leaves of six-week-old switchgrass seedlings of the same accession from which it was collected, except for the Performer switchgrass cultivar due to the lack of seed availability. The Alamo cultivar was used in place of the Performer cultivar. The inoculated seedlings, inside magenta boxes, were placed at 25 °C (ambient temperature) in black trash bags at 100% humidity overnight. After a 24 h incubation, the seedlings were then covered with the magenta box top and placed under grow lights for symptom development at 25 °C. To isolate single urediniospore isolates, newly inoculated seedlings were monitored daily for a well-isolated, non-erumpent pustule on a section of a leaf. The remaining leaves were removed, leaving one leaf with a single pustule. The plants were monitored closely for several days. When the pustule had produced abundant urediniospores, the spores were collected with a small vacuum spore collector. Each of these monospore isolates was separately suspended in 0.001% Tween 20 and used to reinoculate a new seedling of the same plant accession (i.e., cultivar) from which it was collected. Each freshly inoculated seedling was placed back in the growth chamber in a separate magenta box. The spore collection and reinoculation process was repeated at least three times until enough spores (5 mg) were collected for each isolate. Freshly collected spores were desiccated by silica gel for 12 h and stored in Eppendorf tubes at −80 °C.

### 2.3. Virulence of Diverse Puccinia *spp.* Single Spore Isolates

The collected 14 *Puccinia* spp. single-spore isolates were used for inoculation on a set of 38 switchgrass accessions, including 24 genotypes of the SSDP selected to be genetically diverse based on available SNP data [33], as well as 14 genotypes from which the original field collections were made (Appendix A Appendix A). The SSDP contained 372 switchgrass genotypes belonging to 36 accessions [33,37], primarily southern lowland switchgrass (215 genotypes), but it also included some upland genotypes (15 genotypes). Genotypes belonging to the same accession represent individuals from the same cultivar or sampled at the same geographic location. To select the 24 diverse switchgrass accessions from the 372 switchgrass genotypes of the SSDP, Core Hunter v2.0 was used with default weights of 70% of Mean Rogers’ distance and 30 % of Shannon diversity. The analysis was performed based on a total of 11,682 SNPs mined from ~15 Gb of sequence data of 12 genes putatively involved in biomass production [33]. The genotypes were transplanted from the SSDP field panel in Watkinsville, GA, to the greenhouse conditions at UGA for clonal propagation. The inoculation and incubation were performed as previously reported [11]. Briefly, 3–5 mg spores of each isolate were inoculated on three replicates (1 replicate = 1 seedling) of each of the accessions. The disease severity was scored 15 days post-inoculation at the E2 growth stage based on a 1-9 scale developed by other authors [18] and adapted from the wheat rust scale [38]. Average disease scores of <4, 4–5, and >6 were classified as resistant, intermediate, and susceptible reactions, respectively. The virulence pattern of each isolate was assessed based on the switchgrass genotypes showing gene-for-gene responses (major resistance genes involved). These genotypes showed responses of resistance (average disease scores of <4) or susceptibility (average disease scores of <6). The switchgrass genotypes showing intermediate reactions were not considered for race identification.

### 2.4. Genetic Diversity, Genetic Differentiation, and Phylogenetic Analysis 

Fourteen field samples of rust switchgrass collected across seven states in the U.S. were used to assess the genetic diversity of the pathogen population. Genomic DNA was extracted from each sample, as well as from the 14 *Puccinia* spp. single-spore isolates, using the CTAB method [39]. It was checked on a 1% agarose gel for integrity and quality, and used to amplify a region of the *EF-1α* gene based on two primer pairs (first round PCR primer pair: EF1-526F- GTCGTYGTYATYGGHCAYGT and EF1-1567R- ACHGTRCCRATACCACCRATCTT; second round PCR primer pair: EFbasidF- GTGCGGTGGTATCGACAAGC and EFbasidR- CATGTTGTCACCGTGCCATCC) and a nested PCR approach as described by the authors [40]. *EF-1α* has been used for phylogenetic analysis and diversity study of several rust pathogens [40]. PCR was performed on a PTC-gradient cycler (MJ Research). Because urediniospores are dikaryotic, and the two haploid nuclei can be highly heterozygous in rust [41], the EF PCR products were cloned. The PCR products were isolated from a 1.5% agarose gel, purified with an Invitrogen Quick Gel Extraction Kit (Carlsbad, CA, USA), and cloned into the Invitrogen pCR2.1-TOPO cloning vector using the methods described by the manufacturer. PCR was performed with High-Fidelity Taq DNA polymerase to decrease errors associated with PCR amplification. If a single nucleotide change was observed in the sequence of only one clone, it was considered an error and was corrected.

Overall, 720 cloned amplicon sequences (47 clones per field sample and one clone from each of the 14 single-spore isolates), capturing 654 bp of the *EF-1α* gene, were Sanger sequenced by Eurofins. The sequences were aligned under Mega X v10.1.7 [42] and ran under R 3.3.2 [43] for haplotypes (also called multilocus genotype or MLG) identification. To test the resolving power of the SNPs identified in the *EF-1α* gene, a haplotype accumulation curve was generated under R 3.3.2 [43].

In addition, a Principal Coordinates Analysis (PCoA) was performed with the 720 aligned sequences under GenAlEx 6.501 [44] to visualize the structure of the switchgrass rust population in the USA. The population structure in the switchgrass rust was also examined using the Bayesian clustering algorithm STRUCTURE 2.3.4 [45] based on the SNPs identified across the rust haplotypes. STRUCTURE was run for K ranging from 1 to 10, with 10 independent runs for each K, 100,000 burn-in iterations, and 100,000 Markov Chain Monte Carlo (MCMC) repetitions. The optimal number of clusters (K) was identified using the mean posterior probability (ln P(D)) value per cluster and the delta-K method of ln P(D), as defined by [46] under StructureSelector [47]. At the optimal K, haplotypes with membership coefficients of ≥0.8 were assigned to a specific genetic cluster. Furthermore, the phylogenetic relationship between the 720 aligned sequences was performed under Mega X [42] by generating a maximum likelihood (ML) tree using the Tamura-Nei model and 1000 bootstrap replications [48]. Branch support values were estimated using 1000 bootstrap randomizations, and the tree was viewed using FigTree v1.4.4 [49]. The sequences were also used to perform a network analysis using SplitsTree CE 6.0.0 [50].

Furthermore, genetic diversity was analyzed in each rust population sampled within a state and within a genetic cluster. Therefore, genetic diversity indices, such as number of segregating sites (S), number of private SNPs (P), haplotype number (h), haplotype diversity (Hd), nucleotide diversity (π), average number of pairwise nucleotide differences within a population (K), and Watterson’s theta (θ-W per site and θ-W per sequence), were estimated in the *EF-1α* gene, using DnaSP 4.5 Software [51]. The neutrality indices of Tajima’s D [52] and Fu and Li’s D* test [53] in each population were also calculated. The additional pairwise genetic difference between populations was estimated by calculating the gene flow (Nm) and Wright’s F-statistics (Fst) between states and genetic clusters using the 720 aligned sequences of the *EF-1α* gene, under GenAlEx 6.501 [44]. The coefficient of genetic differentiation (Gst) between subpopulations (as defined by STRUCTURE) and its significance were calculated under GenAlEx 6.501 according to the authors [54], formula: Nm = 0.5(1 − Gst)/Gst. Analysis of molecular variance (AMOVA) was also performed using GenAlEx 6.501 [44] in order to investigate the significance of genetic differentiation among and within states and genetic clusters. To provide insight into the reproductive strategy of the pathogen, the index of association (Ia) and standardized index of association (rd^−^) were calculated for each subpopulation using the poppr package in R 3.6.1 [43].

In addition, a stepwise mutation pattern in the *EF-1α* gene was investigated via a phylogenetic analysis (following an ML statistical method and 1000 bootstrap replications under Mega X) to describe the evolution of switchgrass rust in the state of Georgia, where samples were collected in 2011, 2012, 2016, and 2017.

## 3. Results

### 3.1. Virulence Patterns

The virulence patterns of 14 semi-randomly chosen (two from each state) rust isolates were screened on a differential set of 38 switchgrass accessions. Fourteen and eight accessions showed resistance to intermediate and intermediate to susceptible responses, respectively, without a clear virulence differentiation between the rust isolates. Seven accessions showed intermediate reactions for all tested isolates (Appendix A Appendix A).

Only 9 of the 38 switchgrass accessions showed a clear variation in disease response (resistant or susceptible reaction) between the isolates, allowing the identification of eight races (Races 1–8) among the 14 isolates tested (Table 2). Among these nine differential accessions, SNF and PI 315724 presented the same disease response across all the isolates. Race 1 was the most prevalent race and was reported in GA, VA, and TN, while Races 2-6 were only reported once, and Races 7 and 8 were reported two and three times, respectively.

Overall, the eight races differed by 1 to 3 virulence factors. Race 1 was the least virulent race, with virulence recorded only on PI 414066, while Race 3, Race 7, and Race 8 were the most virulent races (three virulence factors). However, races within a state were either not different or differed by only one virulence factor. In fact, only one race was reported in WI (Race 8), GA (Race 1), and OK (Race 7). Two races were observed in each of the following states: TX (Race 5 and Race 8, differentiated on PI 315724/SNF), VA (Race 1 and Race 6 differentiated on PI 414068), TN (Race 1 and Race 4 differentiated on PI 476290), and AL (Race 2 and Race 3 differentiated on PI 315727).

### 3.2. Genetic Diversity

Overall, 97 haplotypes out of the 720 sequences, and 39 segregating sites of the 654 bp of the *EF-1α* gene, were identified (Appendix A Appendix A). The accumulation curve revealed that a large fraction of the diversity in switchgrass rust remains to be discovered (Appendix A Appendix A) and that a more intensive sampling, genotyped with a high-throughput sequencing approach, would likely yield a higher number of haplotypes.

Out of the 97 haplotypes, 55 were represented only once, while 5 were major haplotypes represented more than 29 times on the dataset. These major haplotypes, MLG15, MLG52, MLG91, MLG46, and MLG97, comprised 30, 34, 49, 173, and 181 amplicons, respectively (Table 3). The major haplotypes differed by 11 SNP positions (3 SNPs in coding regions and 8 SNPs in non-coding regions) out of the 39 SNPs identified in the *EF-1α* gene (Table 4). One haplotype (MLG46) was recovered in every state, while MLG91 and MLG97 were recorded in five out of the seven states studied. MLG52 was only found in the five southern states (AL, OK, GA, TN, TX) and totally absent from the two northern states (VA, WI). Across the southern states, MLG97 was the most frequent haplotype in AL, TX, and TN, while MLG46 was the most frequent haplotype in GA and OK. However, in the northern states, two minor haplotypes MLG23 and MLG8 were the most frequent in VA (along with the major haplotype MLG15) and WI, respectively (Table 3, Figure 1, Appendix A Appendix A). 

A genetic diversity study of the switchgrass rust by state showed that Virginia (VA) is one of the most diverse states in the USA, showing the highest average number of nucleotide differences in the *EF-1α* gene (K = 0.87), nucleotide diversity (π = 0.013), θ-W per site (5.633), and θ-W per sequence (0.0086). Georgia (GA) was the least diverse state with the lowest genetic diversity indexes recorded across the seven states studied (Table 5).

Furthermore, rates of gene flow were high between northern states (VA and WI, with Nm = 3.129), and between southern states (AL, GA, OK, TN, and TX; with Nm ranging from 2.182 to 249.5). In contrast, low migration was observed between the northern and southern states (Nm ranging from 0.693 to 2.364). The highest and lowest gene flows were observed between AL and TX and between GA and WI, respectively (Appendix A Appendix A). In addition, the AMOVA on the *EF-1α* gene revealed 14% of the genetic variation is present between states and 86% within states (Appendix A Appendix A).

### 3.3. Amino Acid Changes

Two non-synonymous and non-conservative SNPs (SNP36 and SNP319) and one SNP (SNP571) in intron 3 differentiated MLG15 from all other haplotypes. SNP36 resulted in a Glutamic acid (E) to Glycine (G) substitution at position 45 and, SNP319 in an Isoleucine (I) to Threonine (T) substitution at position 99 in the *EF-1α* protein. I99T (SNP319) is located near known polypeptide binding sites (at positions 21, 64, 67, 68, 70, 72, 74−76, 78, 89, 91, 93−95, 97, 100) (Table 4). The three other non-synonymous and non-conservative amino acid changes E43G (SNP30), D74G (SNP200), and T84I (SNP367) differentiated MLG45, MLG35, MLG36, and MLG78, from all other haplotypes, respectively. D74G is located at a known polypeptide binding site (Appendix A Appendix A).

### 3.4. Genetic Differentiation and Phylogenetic Analysis

Population structure analysis using the SNP data from the *EF-1α* gene revealed that the switchgrass rust population is highly structured in the USA. Two genetic clusters (Pop1 and Pop2) were identified by STRUCTURE (Appendix A Appendix A). No admixed haplotypes were observed. PCoA, phylogenetic, and network analysis also confirmed this genetic subdivision (Figure 2, Figure 3, Appendix A Appendix A). In addition, the migration pattern between the two genetic clusters revealed a low gene flow (Nm = 0.185) and a significant population structure (Gst = 0.730; *p* = 0.001), supporting again this genetic subdivision (Appendix A Appendix A). Pop1 and Pop2 genetic clusters were recorded in every state studied (Appendix A Appendix A).

Genetic clusters 1 (Pop1 in red) and 2 (Pop2 in green) had similar population sizes with 388 and 332 haplotypes, respectively. However, Pop1 was almost twice as diverse as Pop2 in terms of the number of haplotypes H (66 vs. 31), haplotype diversity Hd (0.753 vs. 0.688), average number of nucleotide differences K (2.856 vs. 1.566), nucleotide diversity π (0.004 vs. 0.002), θ-W per site (5.20 vs. 2.977), and θ-W per sequence (0.008 vs. 0.004). A total of 35 and 12 out of the 97 haplotypes were specific to Pop1 and Pop2, respectively (Table 1). MLG15, MLG46, and MLG52 were the major haplotypes in Pop2, while MLG91 and MLG97 belonged to Pop1 (Table 5, Appendix A Appendix A).

A total of 26 out of the 39 SNP positions were polymorphic between the two genetic clusters (frequency range of 1-0.003 within each cluster), while 32 and 18 SNPs were polymorphic within Pop1 and Pop2, respectively. Haplotypes of Pop2 were strictly differentiated from haplotypes of Pop1 by three genetic variations in intron 3 of the *EF-1α* gene (frequency of one within each cluster): deletions of two consecutive nucleotides (SNP535-SNP536), as well as one SNP (SNP538) adjacent to these deletion sites. No SNPs in exon regions differentiated Pop1 and Pop2 (Table 4; Appendix A Appendix A).

To examine the sequence data for departure from neutrality, two neutrality test statistics were used. Tajima’s D test and Fu and Li’s D* test statistics were positive and significant at *p* < 0.05 and *p* < 0.01 for the State of Georgia (D = 2.287) and the State of Wisconsin (D* = 1.642), respectively (Table 3).

In addition, the index of association (Ia) and standardized index of association (rd^−^) for Pop1 and Pop2 were Ia = 2.86, rd^−^ = 0.099 (*p* = 0.01), and Ia = 0.645, rd^−^ = 0.048 (*p* = 0.01), respectively. As expected, this result suggests the existence of significant linkage disequilibrium between the SNPs and a lack of recombination because of the small region analyzed. 

The AMOVA indicated population sub-structuring for the switchgrass rust in the USA, where 73% and 27% of the total observed variation in the *EF-1α* gene were between and within genetic clusters, respectively (Appendix A Appendix A).

### 3.5. Haplotype Evolution in the State of Georgia

A total of 192 out of 720 sequences of the *EF-1α* were obtained from Georgia. Within these 192 sequences, 15 haplotypes of switchgrass rust were recorded between 2011 and 2017; 9 and 6 haplotypes belonged to Pop1 and Pop2, respectively. Four major haplotypes (MLG46, MLG52, MLG91, and MLG97) out of the five reported in the USA were present in Georgia and recorded each year the rust was sampled on the SSDP (in 2011, 2012, 2016, and 2017), with the highest frequencies in 2017 (epidemic year). However, most of the minor haplotypes (9 out of 11) were only present in a single year (Figure 4).

We performed phylogenetic sequence analysis of SNPs from the *EF-1α* gene to investigate how stepwise mutation in the switchgrass rust could give us a better understanding of its local evolution in Georgia and how widespread homoplasy/recombination is among switchgrass rust haplotypes. The evolution of switchgrass rust in Georgia fits very well with a stepwise mutation pattern, where 20 SNPs emerged leading to 15 different haplotypes (Figure 5). As expected, these haplotypes phylogenetically grouped in the two genetic clusters identified (Pop1 and Pop2) and were differentiated by five SNPs located in close proximity (SNP534-536, SNP538, SNP541) in intron 3 of the *EF-1α* gene. However, 2 out of the 20 SNP positions observed in switchgrass rust in Georgia appeared independently in Pop1 and Pop2. These two single nucleotide substitutions, SNP146 (synonymous SNP) and SNP222, were located in exon 2 and intron 2 of the *EF-1α* gene, respectively. In addition, 13 out of the 20 SNPs were rare and were observed in single minor haplotypes (Figure 5).

## 4. Discussion

Switchgrass has been selected as a potential biomass crop for biofuel production. These last two decades, this native species of North America has been researched and bred for increased biomass. With the move from its native habitats to growing areas across the country, it has been under more severe and frequent rust infections. In this study, we assessed the genetic diversity in the *EF-1α* gene of switchgrass rust isolates collected in seven states in the USA, as well as their virulence pattern on a set of 38 diverse switchgrass genotypes.


**Differential lines and SNP markers for race and population identifications**


Nine switchgrass accessions (six upland and three lowland) differentiated the fourteen switchgrass rust isolates into eight distinct races and constituted the first host differentials developed for switchgrass rust race identification. Interestingly, lowland accession SNF from South Carolina and upland accession PI 315724 from Kansas expressed the same phenotypic responses against the isolates tested. It is unknown whether these nine accessions carry different rust resistance genes. To date, rust resistance has not been investigated in switchgrass. The largest feasible gene pool of switchgrass needs to be evaluated against the common races in the USA to identify candidate genes that confer resistance against rust and improve the host differential set for race identification. As more whole-genome sequencing data on switchgrass accessions will be available in the future, identifying the presence of RGH genes could be the first step to improving the host differential set and race identification. A more robust race assessment would also rely on the development of near-isogenic lines of switchgrass carrying single resistance genes and on effector protein profiling of single spore isolates.

In this study, five major switchgrass rust haplotypes were reported, which differed by 11 SNP positions (3 SNPs in coding regions and 8 SNPs in non-coding regions) out of the 39 segregating sites identified in the studied gene. In addition, two genetic clusters (Pop1 and Pop2) were identified in the switchgrass rust population, differentiated by 26 out of the 39 SNP positions in the *EF-1α* gene, including 3 fixed (at a frequency of 1) SNPs in close proximity (SNP535-536, SNP538) in intron 3. From our sequenced ITS region, the rust samples belonged to the *Puccinia novopanici/emaculata* genetic group, but it is unclear if the SNPs are species-specific. Pop1 (35 specific haplotypes; MLG91 and MLG97 predominant haplotypes) and Pop2 (12 specific haplotypes; MLG15, MLG46, and MLG52 predominant haplotypes) exhibited specific haplotypes and distinct predominant haplotypes. Pop1 was almost twice as diverse as Pop2, and the two rust genetic clusters identified were related to very little gene flow. As is the case here, rust populations are typically known to be highly structured worldwide [55].

*EF-1α* codes for a ubiquitous protein involved in the binding of aminoacyl-tRNA to the ribosome during translation. Typically, elongation factors, including *EF-1α*, are highly conserved [56]. However, despite its key role in protein synthesis, mutations in the *EF-1α* gene have been frequently reported in intron sites of various microorganisms, including fungi and nematodes [57]. These sequence variants have proven to be very informative for phylogenetic studies, providing the taxonomic basis of identification for several species [58,59,60], including fungal pathogens [61]. Here, Pop2 haplotypes were differentiated from Pop1 haplotypes by three SNPs located in intron 3. In addition, several non-synonymous SNPs differentiated MLG15 (E45G: SNP36 and I99T: SNP319), MLG45 (E43G: SNP30), MLG35 (D74G: SNP200), and MLG36 and MLG78 (T84I: SNP367) from all other haplotypes in our study. These amino acid changes could lead to structural or functional alteration of the *EF-1α* protein, especially the substitutions (I99T and D74G) identified near or at known binding sites.


**Clonal reproduction and evolution by mutation despite high race and genetic diversities**


Overall, the results across States revealed a high diversity of switchgrass rust based on their virulence patterns with eight races identified among 14 isolates tested. However, one race (Race 1) was predominantly reported (in three states: GA, VA, and TN out of the seven investigated) and races within a state are either not different (GA, OK, and WI) or differed by only one virulence (AL, TN, TX, VA). These findings suggest clonal reproduction of the pathogen. Rust species, including *P. graminis*, *P. triticina,* and *P. striiformis* that cause wheat rusts, are known to quickly adapt to their primary host under strong selection pressure. This adaptation leads to the emergence of new races that break down resistance genes deployed in commercial cultivars, causing epidemic episodes [62,63,64,65]. Clonal evolution of switchgrass rust was also supported by our genetic analysis, where 467 EF 1-α sequences out of 720 grouped into five major haplotypes (MLG15, MLG52, MLG91, MLG46, and MLG97), including one haplotype (MLG46) that was recovered in all seven states. The index of association (Ia) and standardized index of association (rd^−^) were significantly different from 0, revealing linkage disequilibrium between the SNPs and suggesting clonal reproduction of the pathogen. In addition, significant Tajima’s D and Fu and Li’s D* test statistics in Georgia and Wisconsin suggested a positive selection of the switchgrass rust population. In the State of Georgia, particularly, four out of the five major haplotypes were recurrently recorded over multiple years of sampling. Our genetic results also showed that the evolution of switchgrass rust in Georgia fitted very well with a stepwise mutation pattern typical for clonal species, where 20 SNPs successively emerged, leading to 15 different haplotypes. In fact, due to the lack of known sexual recombination in this pathogen, mutations are probably leading to the emergence of new isolates with diverse virulence and genetic profiles, as suggested for several other rust species [66]. However, sexual recombination events creating new genotypes in mostly asexually reproducing pathogenic populations cannot be ruled out [67]. 

Furthermore, 55 out of 97 rust haplotypes were represented only once, supporting the recent rapid expansion of the switchgrass rust population. This finding was also confirmed by the AMOVA, which showed that a high genetic variation within a state explained 86% of the total variation observed in the 720 EF 1-α sequences. One hypothesis that could explain the maintenance of these rare variants in the population is that they provide a particular advantage to the rust population over common haplotypes when environmental conditions are challenging and change rapidly from one season to another. In Georgia, rust infection has been reported in switchgrass since its cultivation expansion as a bioenergy crop and its incidence has increased every year, probably due to an increase in inoculum load. Thirteen out of the twenty SNPs were rare and were observed in single minor haplotypes, supporting this hypothesis of population expansion of switchgrass rust. In addition, minor rust haplotypes were mostly present in only one single year, suggesting evidence of selective sweeps eliminating variants [68] over time in switchgrass rust in Georgia.


**Local adaptation and North–South differentiation despite migration events**


Despite the subdivision of switchgrass rust populations into two distinct genetic clusters, these clusters were recorded in all the states studied, suggesting migration events across states. Migration of genotypes in clonal rust lineages is known to occur over long distances by wind [69]. In the USA, several rust pathogens, including southern rust on corn caused by *P. polysora* [70,71,72], oat and wheat stem rust caused by *P. graminis* [73], wheat stripe rust caused by *Puccinia striiformis* [74], and leaf rust caused by *P. triticina* [75,76], overwinter in southern regions of the USA, and disperse northwards when the temperature rises in spring.

Nevertheless, our study provides genetic evidence for the local adaptation of rust to switchgrass, driven by North–South variation. In fact, MLG52 was only found in the five southern states (AL, OK, GA, TN, TX), being totally absent from the two northern states (VA, WI). MLG97 and MLG46 were the most frequent haplotypes in the southern states of AL, TX, TN, GA, and OK, respectively. However, in the northern states, MLG23, MLG15, and MLG8, were the most frequent haplotypes in VA, and WI, respectively. Furthermore, calculated low levels of gene flow (Nm = 0.693 to 2.364) between the northern (VA and WI) and southern (AL, GA, OK, TN and TX) states compared to calculated high gene flow rates within northern (Nm = 3.129) and southern states (Nm = 2.182 to 249.5) supported this North–South differentiation. Similarly, a recent population genetic analysis using whole genome sequences revealed a northern subpopulation genetically distinct from a southern subpopulation in *P. novopanici* in the USA [23]. Geographic barriers [77,78] and host selection [63,79,80] could explain the geographical differentiation and local adaptation of switchgrass rust. Several other climatic factors were also reported to affect rust species’ evolution and local adaptation, including temperature [81,82]. Northern and southern USA regions are characterized by cold winters and upland switchgrass ecotypes, and warm winters and lowland ecotypes, respectively. Specialization to a particular switchgrass ecotype could have contributed to the niche separation between northern and southern rust genotypes, as observed for other fungal pathogens. To date, it is not clear where and to what extent the host plays a role in switchgrass rust adaptation. It is also unknown whether temperature or any other climate variables have impacted switchgrass rust differentiation. Genetic incompatibilities and reproductive isolation could also explain the coexistence of multiple genetic subpopulations in sympatry and the maintenance of population structure in fungal pathogenic species [83]. 

In addition, besides the evidence of local adaptation, rust switchgrass exhibited parallel genomic adaptation events. In fact, the phylogenetic analysis suggested that two (SNP433 and SNP509) out of twenty SNP positions of the *EF-1α* gene observed in switchgrass rust haplotypes in Georgia appeared independently in Pop1 and Pop2. The *EF-1α* and virulence profiles of the 14 single-spore isolates also showed convergent evolution, where isolates of Race7 did not share a common ancestral origin but probably arose independently in Pop1 and Pop2. Signatures of convergent evolution and homoplasy events have been suggested to occur in several rust pathogens [84,85]. An alternative hypothesis could be that rare genetic recombination episodes occur in the switchgrass rust population. 

Interestingly, northern states showed a high genetic diversity of switchgrass rust compared to southern states. In fact, Virginia and Georgia showed the highest and lowest number of races and genetic diversity indexes, respectively. Northern states could be key regions for switchgrass rust migration routes, where the pathogen is sexually reproducing and acquiring the genetic variation necessary for its adaptive evolution under diverse environmental conditions. Although neither alternative hosts nor basidiospores, pycniospores, or aeciospores have been identified for switchgrass rust, the production of teliospores, spores that initiate the sexual cycle of rust fungi, have been reported in northern isolates at the end of the growth season under field conditions [12]. As observed for other rust fungi, the telia formation in northern state isolates could be related to many factors, including temperature [86,87], race [88], and host cultivars [87,89]. In the USA, variation in telia formation of *P. striiformis* was affected by various factors, including region-year interaction [90]. In this study, the reticulated evolutionary process revealed by the network analysis of the *EF-1α* gene showed potential recombination events in *Puccinia* spp. More studies are needed on the natural telia formation rate and the roles of teliospores and potential alternate hosts in the life cycle of switchgrass rust. 

## 5. Conclusions

In this study, the pathogenicity responses of 14 single-spore rust isolates on a set of 38 genetically diverse switchgrass genotypes and SNP variations in 720 sequences of the *EF-1α* gene from 14 field samples collected across seven U.S. states were assessed. The findings suggested (1) clonal reproduction of the pathogen and its evolution by mutation despite high race and genetic diversity, and (2) local adaptation of switchgrass rust populations despite their migration across the studied states.

## Figures and Tables

**Figure 1 pathogens-14-00194-f001:**
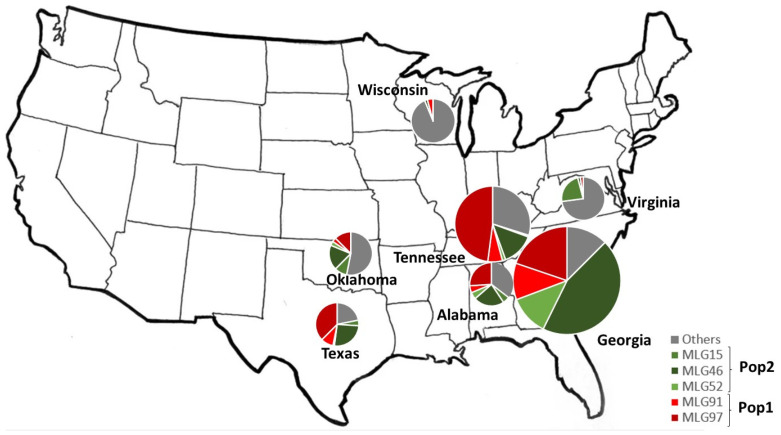
Geographic location of the five major haplotypes of switchgrass rust populations in seven U.S. states. Pie charts are at scale, with the exception of Virginia and Wisconsin, which are 50% larger.

**Figure 2 pathogens-14-00194-f002:**
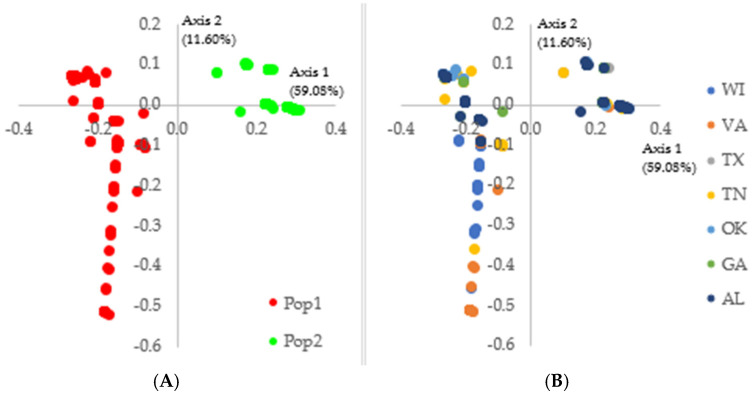
Principal Coordinates Analysis of the 720 *Puccinia* spp. haplotypes from switchgrass based on 39 SNPs in the elongation factor 1-α gene. The haplotypes are color-coded based on: (**A**) their affiliation to STRUCTURE genetic clusters at K  =  2; and (**B**) their state of origin. Pop1 and Pop2 are in red and green, respectively. WI, VA, TX, TN, OK, GA, AL represents the States of Wisconsin, Virginia, Texas, Tennessee, Oklahoma, Georgia, and Alabama, respectively.

**Figure 3 pathogens-14-00194-f003:**
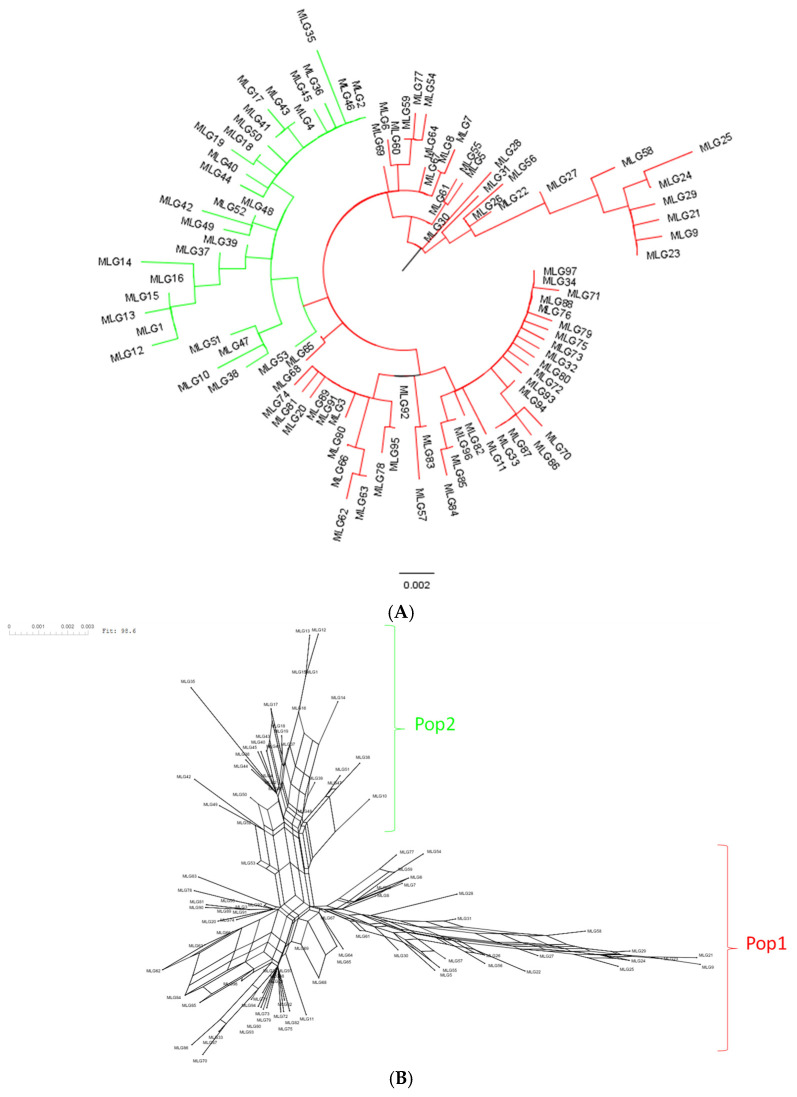
Phylogenetic relationships between 97 *Puccinia* spp. haplotypes from switchgrass based on 39 SNPs in the elongation factor 1-α gene: (**A**) Maximum likelihood tree with red and green branches, indicating Pop1 and Pop2 genetic clusters, respectively; (**B**) Network analysis showing reticulated evolution.

**Figure 4 pathogens-14-00194-f004:**
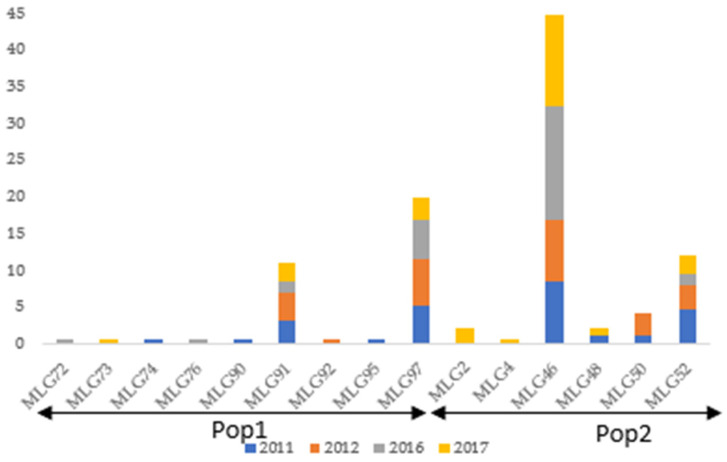
Haplotype frequencies by year of 192 *Puccinia* spp. haplotypes from switchgrass based on 39 SNPs in the elongation factor 1-α gene, recorded in the State of Georgia.

**Figure 5 pathogens-14-00194-f005:**
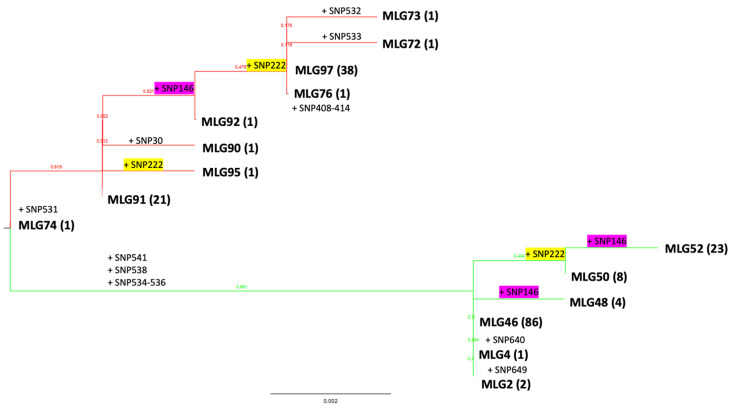
Maximum likelihood tree showing the emergence of new switchgrass rust haplotypes by step mutations in the elongation factor 1-α gene in the State of Georgia. Mutations are indicated by ‘+’ sign in the branches followed by the SNP position. SNP222 (highlighted in yellow) and SNP146 (highlighted in pink) mutations appeared independently in Pop1 (red) and Pop2 (green) genetic clusters. The number of times a haplotype was present in the GA dataset is indicated in parenthesis.

**Table 1 pathogens-14-00194-t001:** Fourteen *Puccinia* spp. field isolates collected and their switchgrass field and accession origin.

Isolate ID ^1^	State ^2^	Switchgrass Genotype ^3^
AL-3539-1-05	Alabama	PI 315728 (L)
AL-WILD-1-02	Alabama	PI 421999 (L)
GA2011A03	Georgia	PI 422006 (Alamo) (L)
GA2012G04	Georgia	PI (422001) (Stuart) (C)
OK15x4-6-05	Oklahoma	PI 476290 (L)
OK13X12-4-07	Oklahoma	PI 476294 (U)
TN1B-3-02	Tennessee	Performer (L)
TNP-4-01	Tennessee	PI 422006 (Alamo) (L)
TX39-6-06	Texas	PI 537588 (Dacotah) (U)
TX172-2-01	Texas	PI 642191 (Summer) (U)
VA2015G04	Virginia	PI 469228 (Cave-in-Rock) (U)
VA2015E05	Virginia	PI 315724 (U)
WI2015B03	Wisconsin	PI 469228 (Cave-in-Rock) (U)
WI2015D04	Wisconsin	PI 422006 (Alamo) (L)

^1^ Identification number given to isolates after collection. First two letters are the two-letter state abbreviation corresponding to the state of isolate collection. Remaining ID corresponds to either the year of collection or field location. ^2^ State *Puccinia* spp. field isolates were collected from. ^3^ Switchgrass genotypes *Puccinia* spp. field isolates were collected from. Ecotypes were added in parenthesis [30,36].

**Table 2 pathogens-14-00194-t002:** Virulence patterns and race identification of 14 *Puccinia* spp. isolates from switchgrass assessed on the eight switchgrass accessions showing differential disease responses.

	Accession ^1^							
Isolate ID_Race#_MLG (Pop) ^2^	PI 315725 (U; MS)	PI 414068 (U; KS)	PI 476290 (L; NC)	SNF (L; SC)/ PI 315724 (U; KS)	PI 421520 (U; OK)	PI 315727 (L; NC)	PI 414067 (U; NC)	PI 414066 (U; NM)
TN1B-3-02_Race1_MLG97 (Pop1)	R	R	R	R	R	R	R	S
GA2012G04_Race1_MLG97 (Pop1)	R	R	R	R	R	R	R	S
GA2011A03_Race1_MLG91 (Pop1)	R	R	R	R	R	R	I	S
VA2015G04_Race1_MLG23 (Pop1)	R	R	R	R	R	R	I	S
AL-3539-1-05_Race2_MLG97 (Pop1)	R	R	R	R	R	R	S	S
AL-WILD-1-02_Race3_MLG97 (Pop1)	R	R	R	R	R	S	S	S
TNP-4-01_Race4_MLG97 (Pop1)	R	R	S	R	R	R	R	S
TX172-2-01_Race5_MLG97 (Pop1)	R	R	S	R	S	R	R	I
VA2015E05_Race6_MLG64 (Pop1)	R	S	R	R	R	R	I	S
OK15x4-6-05_Race7_MLG85 (Pop1)	S	S	I	R	R	I	R	S
OK13X12-4-07_Race7_MLG42 (Pop2)	S	S	I	R	R	I	R	S
TX39-6-06_Race8_MLG97 (Pop1)	R	R	S	S	S	R	R	I
WI2015D04_Race8_MLG67 (Pop1)	R	R	S	S	S	I	R	R
WI2015B03_Race8_MLG22 (Pop1)	R	R	S	S	S	I	R	R

^1^ Disease was evaluated on a 1–9 scale and converted to resistant (<4), intermediate (4–5), and susceptible (>6) reactions. In parenthesis are the ecotype (U: upland; L: lowland) followed by the state of origin for each accession. ^2^ MLG: haplotype (multilocus genotype) ID identified based on the combination of SNPs across the 654 bp of the *EF-1α* gene; Pop: Population affiliation (Pop1, Pop2) identified in the *EF-1α* gene using the Bayesian clustering algorithm STRUCTURE 2.3.4.

**Table 3 pathogens-14-00194-t003:** Number of amplicons recorded by the state for the five major haplotypes identified in the elongation factor 1-α gene in the switchgrass rust population in the USA.

Haplotype ID	Genetic Cluster	AL	GA	OK	TN	TX	VA	WI	Total
MLG15	Pop2	5	0	9	1	4	11	0	30
MLG46	Pop2	22	86	18	20	25	1	1	173
MLG52	Pop2	5	23	3	2	1	0	0	34
MLG91	Pop1	5	21	3	9	9	0	2	49
MLG97	Pop1	25	38	12	69	36	1	0	181

AL: Alabama; GA: Georgia; OK: Oklahoma; TN: Tennessee; TX: Texas; VA: Virginia; WI: Wisconsin; Total: number of times the haplotype was recorded in the dataset.

**Table 4 pathogens-14-00194-t004:** Single nucleotide polymorphism (SNP) positions in the elongation factor 1-α gene detected between the five major haplotypes reported in the switchgrass rust population in the USA.

Haplotype ID (Genetic Cluster)	Total	SNP36/AA45	SNP146/AA60	*SNP222*	SNP319/AA99	*SNP531*	*SNP534*	*SNP535*	*SNP536*	*SNP538*	*SNP541*	*SNP571*
MLG15 (Pop2)	30	**G (G)**	G (L)	*T*	**C (T)**	*T*	*-*	*-*	*-*	*A*	*C*	*A*
MLG46 (Pop2)	173	**A (E)**	G (L)	*T*	**T (I)**	*T*	*-*	*-*	*-*	*A*	*C*	*T*
MLG52 (Pop2)	34	**A (E)**	T (L)	*C*	**T (I)**	*T*	*-*	*-*	*-*	*A*	*C*	*T*
MLG91 (Pop1)	49	**A (E)**	G (L)	T	**T (I)**	*C*	*T*	*T*	*G*	*G*	*T*	*T*
MLG97 (Pop1)	181	**A (E)**	T (L)	C	**T (I)**	*C*	*T*	*T*	*G*	*G*	*T*	*T*

Total: number of times the haplotype was recorded in the dataset; SNP position/Amino Acid (AA) position are indicated: SNPs in italics are located in intronic regions of the elongation factor 1-α gene; SNPs in bold are non-synonymous SNP; Amino acids are indicated in parentheses.

**Table 5 pathogens-14-00194-t005:** Nucleotide/haplotype diversity and neutrality tests of (A) the two genetic clusters and, (B) the seven states using 720 *Puccinia* spp. haplotypes from switchgrass based on 39 SNPs in the elongation factor 1-α gene.

	Pop	Total	S	Private SNP	H	Hd ± SD	K	π	θ-W per Sequence	θ-W per Site	D	Fu and Li’s D* Test Statistic
**A**	Pop1	388	32	35 (1–0.003)	66 (66)	0.753 (0.00049)	2.856	0.004	5.201	0.008	−1.223	−0.723
	Pop2	332	18	12 (1–0.003)	31 (31)	0.688 (0.00070)	1.566	0.002	2.977	0.005	−1.197	−1.711
**B**	AL	96	19	4 (0.01–0.021)	25 (9)	0.864 (0.00052)	4.899	0.007	3.894	0.006	0.751	0.412
	GA	192	11	11 (0.005–0.021)	15 (7)	0.709 (0.00066)	3.686	0.006	1.886	0.003	2.287 *	−0.807
	OK	96	18	5 (0.01–0.042)	24 (10)	0.907 (0.00017)	5.886	0.009	3.894	0.006	1.489	0.853
	TN	144	25	2 (0.007–0.007)	33 (13)	0.744 (0.00127)	4.125	0.006	4.870	0.007	−0.442	−1.998
	TX	96	15	1 (0.01)	18 (2)	0.784 (0.00091)	4.59	0.007	2.920	0.004	1.591	−0.026
	VA	48	24	1 (0.021)	20 (11)	0.87 (0.000830	8.496	0.013	5.633	0.008	1.672	0.843
	WI	48	18	3 (0.104–0.313)	23 (18)	0.932 (0.00048)	4.494	0.007	4.056	0.006	0.342	1.642 **
	Total	720	39		97	0.862 (0.00007)	5.372	0.008	5.870	0.009	−0.225	0.728

Total: Number of sequences assessed; S: Number of polymorphic sites; Private SNP: Number of private SNPs (frequency range); H: Number of haplotypes (number of specific haplotypes); Hd: Haplotype diversity (SD: Standard deviation); K: Average number of pairwise nucleotide differences; π: Nucleotide diversity; θ-W: Watterson’s theta; D: Tajima’s D test statistics. *: *p* < 0.05; **: *p* < 0.02.

## Data Availability

The original contributions presented in this study are included in the article/Appendix A. Further inquiries can be directed to the corresponding authors.

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
