# Peer review of "Virulence and Genetic Diversity of Puccinia spp., Causal Agents of Rust on Switchgrass (Panicum virgatum L.) in the USA"

_pathogens, 2025, doi:10.3390/pathogens14020194_

Round 1

Reviewer 1 Report

Comments and Suggestions for Authors

The authors have meticulously performed the experiments and presented them nicely. But, a few points need to be clarified before publishing this manuscript:

1. Information of resistance gene homologues (RGH) present in eight differentials used for race identification of Puccinia spp. infecting  switchgrass is essential

2. How do mutation/SNPs in EF-Tu genes lead to the evolution of new races of Puccinia spp.? Please clarify.

3. Why are not authors focusing on variation in effector proteins/ effector profiling for defining race concept

Author Response

Responses to Reviewer 1:

The authors have meticulously performed the experiments and presented them nicely. But, a few points need to be clarified before publishing this manuscript:

  1. Information of resistance gene homologues (RGH) present in eight differentials used for race identification of Puccinia spp. infecting switchgrass is essential

Response: We currently don’t have information about the presence/absence of the RGH genes in the host differential set used. We thank the review for pointing out the value of this information and we added a few lines regarding this fact in the Discussion section (Lines 473-476, page 14).

  1. How do mutation/SNPs in EF-Tu genes lead to the evolution of new races of Puccinia spp.? Please clarify.

Response: Mutation is the main evolutionary force explaining the emergence of new races in Puccinia spp. Because the pathogen is mostly evolving clonally, mutations in the genome are in linkage. The mutation in EF gene and ITS region don’t lead to the evolution of new races of Puccinia spp., but the SNPs in those genes and the SNPs in genes leading to new races are most likely in linkage. However, the EF and virulence profiles of the 14 single spore isolates (Table 2) also showed convergence evolution where isolates of Race7 do not have a common ancestral origin but probably arose independently in Pop1 and Pop2. This point has been clarified in the Discussion section (Lines 584-587, page 16).

  1. Why are not authors focusing on variation in effector proteins/ effector profiling for defining race concept

Response: Currently, there is no effector proteins identified for switchgrass rusts. For now, using a host differential set remains a good way to identify races. As more genomic data will be available on Puccinia spp. of switchgrass, we will be able to more accurately provide effector protein profiling. This point has been clarified in the Discussion section (Lines 476-478, page 14).

Comments as per the marked-up PDF

Italicize “Puccinia” in the headers

Response: We believe that “Puccinia” should not be italicized since the header is in italics.

Table 3: Specify the full name of the states in foot note; Arrange these values properly

Response: The full name of the States is now specified in foot note and the values are properly arranged in the Table.

Reviewer 2 Report

Comments and Suggestions for Authors

This paper describes important studies on the virulence and genetic diversity of Puccinia spp. on switchgrass (Panicum virgatum L.) in the United States. The review provides a good overview of the main rust fungi pathogens on this crop, describes the species-specific variability of rust fungi, and the primers used for these studies. Several studies on the genetic basis of rust resistance in switchgrass are presented. The materials and methods are described in sufficient detail, with the exception of a few points described in the comments below. Very interesting and significant results were obtained. As a result, genetic diversity in the EF-1α gene of switchgrass rust isolates collected in seven US states was assessed; virulence of 14 isolates on a set of 38 different switchgrass genotypes was described. There are some questions and comments:

1. The chapter on the review describes well the types of rust pathogens that can affect millet crops. There is not enough information on the virulence and existing races of isolates of the pathogen common on millet in the regions where this crop is grown.

2. In the chapter on materials and methods, please provide a link or explanation of who proposed the differential set of 38 millet samples to study the virulence of isolates. And by what method and how were the races assigned to the isolates. Please provide a more detailed description.

3. In Table 3, the last column can be signed - total.

4. In the text, when describing the results, the authors use the term rust populations, population structure .. (for example, line 350, 351, etc.). Considering that only 14 isolates were studied, maybe it is better to describe the sample of isolates studied? To what extent is it correct to talk about a population in this case?

Author Response

Responses to Reviewer 2:

This paper describes important studies on the virulence and genetic diversity of Puccinia spp. on switchgrass (Panicum virgatum L.) in the United States. The review provides a good overview of the main rust fungi pathogens on this crop, describes the species-specific variability of rust fungi, and the primers used for these studies. Several studies on the genetic basis of rust resistance in switchgrass are presented. The materials and methods are described in sufficient detail, with the exception of a few points described in the comments below. Very interesting and significant results were obtained. As a result, genetic diversity in the EF-1α gene of switchgrass rust isolates collected in seven US states was assessed; virulence of 14 isolates on a set of 38 different switchgrass genotypes was described. There are some questions and comments:

  1. The chapter on the review describes well the types of rust pathogens that can affect millet crops. There is not enough information on the virulence and existing races of isolates of the pathogen common on millet in the regions where this crop is grown.

Response: This is the first study regarding the virulence spectrum of switchgrass rust.

  1. In the chapter on materials and methods, please provide a link or explanation of who proposed the differential set of 38 millet samples to study the virulence of isolates. And by what method and how were the races assigned to the isolates. Please provide a more detailed description.

Response: The information related to the differential set of 38 switchgrass accessions is already detailed in the M&M section (under 2.3).

The section reads as follow: “The 38 switchgrass accessions included 24 genotypes of the SSDP selected to be genetically diverse based on available SNP data, as well as 14 genotypes from which the original field collections were made”… “To select the 24 diverse switchgrass accessions from the 372 switchgrass genotypes of the SSDP panel, Core Hunter v2.0 was used with default weights of 70% of Mean Rogers’ distance and 30% of Shannon diversity. The analysis was performed based on a total of 11,682 SNPs mined from ~ 15 Gb of sequence data of 12 genes putatively involved in biomass production [33]”.

  1. In Table 3, the last column can be signed - total.

Response: The last column of Table 3 is now signed as “Total”. This was also modified in Table 4 and Table 5 for consistency.

  1. In the text, when describing the results, the authors use the term rust populations, population structure .. (for example, line 350, 351, etc.). Considering that only 14 isolates were studied, maybe it is better to describe the sample of isolates studied? To what extent is it correct to talk about a population in this case?

Response: Population structure was assessed on genetic data based on 720 haplotypes sequences (47 clones per field sample and one clone from each of the 14 single-spore isolates) derived from 14 fields of switchgrass across seven states in the U.S. This was corrected in the M&M section (Lines 192-194, 210-211, page 5).

Reviewer 3 Report

Comments and Suggestions for Authors

The manuscript describes characterization of virulence and genetic diversity of a rust population in switch grasses. The finding provides first-hand information about the population diversity of the fungal pathogen in US, which is important for breeding and deployment of resistant accessions of the crop. No major concern was identified, but a few comments for authors to consider.

Any research work has been done to investigate rust pathogen diversity?

Line 126: from a single pustule?

Line 144: what is talc?

Line 166-169 May want to explain more about accessions and genotype in switchgrass? Switchgrass is an outcrossing plant species. How to maintain the pure genetic line for genetic analysis?

Line 183 what are clear response of resistance and susceptibility? Please describe

Line 188 14 seems to too low for a population diversity study. Nee to justify with this small number of isolates.

Line 192 why EF1 was chose for diversity study?

Table 2, what is MLG and pop for those isolates, please explain and denote under the table.

Any connection between race classification and genetic classification? This seems not be discussed.

Space between number and units across the manuscript

Author Response

Responses to Reviewer 3:

The manuscript describes characterization of virulence and genetic
diversity of a rust population in switch grasses. The finding provides
first-hand information about the population diversity of the fungal
pathogen in US, which is important for breeding and deployment of
resistant accessions of the crop. No major concern was identified, but a
few comments for authors to consider.

Any research work has been done to investigate rust pathogen diversity?
Response: Rust pathogen diversity on switchgrass in the U.S. has been investigated at the species level based on rDNA sequencing. This is specified in the Introduction section (Line 62-68, page 2). In our study, the rust samples were belonging to Puccinia novopanici/emaculata genetic group (Supplementary Data Figure S1) (Lines 129-130, page 3).

Line 126: from a single pustule?
Response: Spores were not collected from a single pustule.  

Line 144: what is talc?

Response: Rust spores are usually mixed with talcum powder for inoculation. It is a mineral made of magnesium, silicon and oxygen with the following molecular formula: 3MgO-4SiO2-H2O. This have been added to the manuscript.

Line 166-169 May want to explain more about accessions and genotype in
switchgrass? Switchgrass is an outcrossing plant species. How to
maintain the pure genetic line for genetic analysis?

Response: Yes, switchgrass is an outcrossing species. Genotypes belonging to the same accession represent individuals from the same cultivar or sampled at the same geographic location. This was clarified in the manuscript (Lines 170-172, page 4). Switchgrass is a perennial grass; pure genetic lines are maintained in the field or greenhouse, and easily clonally propagated through tiller division for genetic analysis.

Line 183 what are clear response of resistance and susceptibility?
Please describe
Response: This was clarified in the manuscript and the revised text reads as follow: “The virulence pattern of each isolate was assessed based on the switchgrass genotypes showing gene-for-gene responses (major resistance genes involved): These genotypes showed responses of resistance (average disease scores of <4) or susceptibility (average disease scores of <6). The switchgrass genotypes showing intermediate reactions were not considered for race identification”.

Line 188 14 seems to too low for a population diversity study. Nee to
justify with this small number of isolates.

Response: We actually used 720 cloned amplicon sequences for the diversity study: 47 clones per field sample (for 14 fields) and one clone from each of the 14 single-spore isolates. This was clarified in the revised manuscript.

Line 192 why EF1 was chose for diversity study?

Response: EF1 has been used for fungal diversity study in several published works, including for rust pathogens (van der Merwe et al., 2007, O’Donnell et al., 2008). This was added to the manuscript (Line 196-197, page 5).

Table 2, what is MLG and pop for those isolates, please explain and
denote under the table.

Response: MLG refers to the haplotype (Multilocus genotype) ID. MLG was identified based on the combination of SNPs across the 654 bp of the EF-1α gene. This was added in Line 209. Pop refers to the population affiliation (Pop1, Pop2) identified in the EF-1α gene using the Bayesian clustering algorithm STRUCTURE 2.3.4. MLG and Pop were further denoted and explained in the footnote of Table.

Any connection between race classification and genetic classification?
This seems not be discussed.

Response: From our dataset it is hard to give any conclusion about the correction between race (virulence profiles) and the genetic classification (Population cluster based on SNPs of the EF-1α gene) because 13 out of the 14 single-spore isolates used for race identification belonged to Pop1 genetic cluster; Only 1 isolate (OK13X12-4-07_Race7_MLG42) belonged to Pop2. However, a comment was added in the discussion about the convergence in virulence profile within both genetic groups (Lines 584-587, page 16).

Space between number and units across the manuscript

Response: This was corrected across the manuscript.

Round 2

Reviewer 1 Report

Comments and Suggestions for Authors

Justification is accepted